# Potential Influences of Bacterial Cell Surfaces and Nano-Sized Cell Fragments on Struvite Biomineralization

**Robert J. C. McLean *** and **Erin T. Brown**

Department of Biology, Texas State University, 601 University Drive, San Marcos, TX 78666, USA; etb35@txstate.edu
* Correspondence: McLean@txstate.edu; Tel.: +1-512-245-3365

**Abstract:** Struvite ($MgNH_4PO_4 \cdot 6H_2O$) calculi are formed as a result of urinary tract infections by *Proteus mirabilis* and other urease-producing bacteria. During struvite formation, the bacteria grow as biofilms, and thus crystals are formed in close association with bacterial cell surfaces and biofilm matrix components. Small nano-sized objects (originally termed "nanobacteria") have been described in association with urinary calculi including struvite calculi. A much more likely explanation of these nano-structures is outer membrane vesicles (OMVs) that can be produced by *P. mirabilis* and other Gram-negative bacteria. In this brief review, we describe the association of bacterial cell surfaces and biofilm matrix components with metal binding and the generation of chemical microenvironments during struvite formation; we propose potential mechanisms whereby OMVs can influence struvite crystal growth and biomineralization.

**Keywords:** bacterial cell surfaces; polysaccharides; nanobacteria; outer membrane vesicles; lipopolysaccharide; Proteus mirabilis; biofilm; microenvironment

---

## 1. Introduction

The roles of microorganisms in mineral formation and the associated interdisciplinary work in microbiology and geology are becoming more recognized. There are several journals that are devoted to this subject, (e.g., Biogeosciences, Geobiology and the Geomicrobiology Journal), in addition to publications that appear in a range of other journals. Microbial-associated biomineralization is also an aspect of some infections in humans and other animals. Notable examples include dental calculus which is a mineral associated with dental plaque [1] and infectious urinary calculi [2], associated with urinary tract infections (UTIs) in which the dominant mineral is struvite ($MgNH_4PO_4 \cdot 6H_2O$). This mineral was named for Heinrich Christian Gottfried von Struve (Baron von Struve, 1772–1851) [3].

Urinary calculus disease (urolithiasis) affects an estimated 600,000 patients per year in the US [4]. The majority of urinary calculi (~80%) contain calcium minerals including calcium oxalate and calcium phosphate. Mg-containing struvite urinary calculi account for approximately 10–15% of all urinary calculi and are especially problematic in patients with indwelling urinary catheters and the associated biofilm infections [5,6]. Struvite calculi are also associated with urinary infections in some domestic animals, notably dogs and cats, and this mineral has been also found in bat guano [3]. Traditionally, the majority of urinary calculi having the dominant minerals of calcium oxalate and calcium phosphate were considered to not be associated with UTIs, although a recent culture-independent study of the urinary calculus microbiome raises questions on that assumption [7].

One common feature of struvite urinary calculi is their association with urease-producing bacteria (reviewed in [8,9]). The prominent urease-producing species associated with UTIs is *Proteus mirabilis*.

However, other urease-producing species have been described, including other *Proteus* spp., and the genera *Providencia, Morganella*, and *Staphylococcus*. In 1976, Donald Griffith and colleagues proposed that urease represented a key virulence factor [10], and this work was confirmed by Harry Mobley and colleagues [11,12] who constructed a deletion mutant in the *ureC* gene in *P. mirabilis* and observed reduced virulence and a lack of struvite formation by the *ureC* mutants in an animal model of ascending UTI. The major nitrogen-containing molecule in urine is urea [8], and urease (EC: 3.5.1.5) catalyzes the hydrolysis of urea as follows:

$$NH_2CONH_2 + H_2O \rightarrow 2NH_3 + CO_2$$

At pH < 9, ammonia ($NH_3$) reacts with water to form ammonium ($NH_4^+$) with a resulting increase in pH:

$$NH_3 + H_2O \rightarrow NH_4^+ + OH^- \rightarrow pH\uparrow$$

Given the chemical composition of urine, $Mg^{2+}$ precipitates as struvite ($MgNH_4PO_4 \cdot 6H_2O$) and $Ca^{2+}$ precipitates as carbonate apatite ($Ca_{10}(PO_4)_6CO_3$), both of which are poorly soluble under alkaline conditions [8]. Struvite is acid labile, and depending upon urine chemistry, will dissolve below pH 7 [13].

In addition to *ureC*, other genes have also been identified as essential in *P. mirabilis* UTIs [14]. Many of these genes are associated with nutrient acquisition, adhesion to urinary epithelial cells, altered metabolism in the urinary tract environment or evading the immune response. Their role in *P. mirabilis* growth and survival during infection is understandable in that context [15].

One notable feature of struvite calculi is the presence of bacterial biofilms in association with struvite calculi [16]. Biofilms, defined as surface-adherent microbial communities, represent a common growth strategy of bacteria in many infections including struvite urolithiasis [17]. In this review, we address some features of biofilms, and biofilm components including nano-sized structures [18] that may contribute to struvite mineral formation.

## 2. Bacterial Cell Surface Structures, Nano-Sized Objects, and Biofilms

A number of bacteria have been associated with struvite growth. The common feature of these organisms is urease. While the most prominent organisms identified are *P. mirabilis*, a Gram-negative member of the family *Morganellaceae* [19], urease activity has been described in many other bacteria including other members of the *Morganellaceae* and the *Enterobacteriaceae*. Gram-positive *Staphylococcus* spp. and *Ureaplasma urealyticum*, an organism that lacks a cell wall, have also been implicated [20]. Given the mixed microbial populations in the urinary tract, it is not surprising that other organisms have been shown to contribute to *P. mirabilis* virulence [21]. Scanning electron microscopy (SEM) and some transmission electron microscopy (TEM) studies of struvite calculi from clinical specimens [22,23], animal studies [24], and in vitro lab investigations [25,26] illustrate the close association of struvite crystals within bacterial biofilms. Small structures, originally identified as nanobacteria, have also been described [27] in SEM and TEM investigations. We now address some features of bacterial cell surfaces, including nano-sized objects, and their role in biofilm formation and function.

### 2.1. Nanobacteria and Nano-Sized Objects

In 1998, Kajander and Çifçioglu described the association of nanometer-scale objects (50–200 nm diameter) that were referred to as nanobacteria [27] and proposed these structures as potential causative agents of urinary calculi. In this work, they also were able to extract and sequence the DNA from these nanobacteria. Nanobacteria (also spelled nannobacteria) had been described by Robert L. Folk in association with travertine [28], a calcium carbonate rock formed by cyanobacteria [29]. Except for their small size (50–200 nm), these objects greatly resembled spherical bacteria (coccus shape) when examined by SEM. Subsequent studies by Folk and other investigators (reviewed in [30]) described

these structures in many different environments, and one notable study of a meteorite used nano-sized objects, interpreted as nanobacteria, as evidence of past life on Mars [31].

Several studies were conducted largely as a result of both the nano-sized objects associated with urinary calculi [27], and the McKay et al. investigation of potential extraterrestrial life [31]. In one report on the size limits of independent life by the USA National Academy of Science, calculations based on essential proteins, nucleotides and membranes, placed an estimated minimum size for an independent lifeform as a sphere of 250 ± 50 nm [32], which is within the size range of nanobacteria (50–200 nm) [30]. A study by Cisar et al [33] showed that nano-sized aggregates could form from the association of organic matter and very small microcrystals. In addition, the DNA isolated and sequenced in the original study [27] was identical to a common PCR contaminant. A study by Kirkland et al. [34] showed that bacterial cellular debris from phage-induced lysis, as well as the aggregation of polysaccharides with $Ca^{2+}$, could produce nano-sized objects resembling nanobacteria. Another potential structure would be outer membrane vesicles (OMVs) that are produced by many Gram-negative bacteria [35]. Finally, one notable observation is that nanobacteria have been described in environments in which normal sized bacteria also reside [30]. Given these issues, it is much more likely that nano-sized objects < 250 nm diameter are microbial fragments or small aggregates of minerals and organic matter, rather than independent lifeforms [36].

### 2.2. Nano-Sized Outer Membrane Vesicles

OMVs are nano-sized mostly spherical structures (diameter of 20–250 nm) [37] and have been shown to be present in biofilm matrices in both environmental [38] and medical [39] situations. OMVs do contain a number of organic molecules (reviewed in [39,40]) including peptides, nucleic acids, and certainly the surrounding membrane component consisting of lipopolysaccharide (LPS) from the Gram-negative outer membrane. OMV formation has been associated with a number of features (reviewed in [35,40]) including microbial competition, predation, genetic exchange, and interspecies communication. In some organisms, notably of the genus *Neisseria*, OMV formation is a key virulence attribute as the LPS-containing OMVs along with associated virulence factors cause tissue inflammation when they attach to host tissue [41] adjacent to the infecting organisms. In the genus *Neisseria*, the outer membrane is referred to as lipooligosaccharide (LOS), rather than LPS [41,42].

LPS is an amphipathic molecule located on the external portion of the outer membrane (reviewed in [42]). LPS has a conserved hydrophobic inner region consisting of lipid A containing a bis-phosphorylated glucosamine disaccharide bound to several acyl chains. The acyl chains bind LPS via hydrophobic interactions to the phospholipids in the inner leaflet of the outer membrane. The hydrophilic region consists of a core region containing phosphate, and one or more sugar acids, which provide the hydrophilic character. O antigens, representing repeating polysaccharide units, are present in many LPS molecules. The chemistry of O antigens is strain specific, and in many Gram-negative bacteria such as *Pseudomonas aeruginosa*, the O antigen layer can extend up to 10 nm from the cell surface [43]. While OMVs are produced by many organisms, the mechanism of formation is best understood in *P. aeruginosa*.

Many bacteria sense and respond to population density through a mechanism known as quorum sensing (QS) (reviewed in [44]). *P. aeruginosa* has several QS systems, one of which is mediated by the Pseudomonas quinolone signal (PQS) (2-heptyl-3-hydroxyl-4-quinolone) [45]. PQS binds directly to LPS on the outer membrane of *P. aeruginosa* and induces OMV formation [46,47]. PQS is also able to induce OMV formation in several other Gram-negative bacteria [35]. Culture growth conditions have been shown to influence OMV formation in *P. aeruginosa* [46]. To the best of our knowledge, no work has been performed to investigate the role of urine chemistry on OMV formation.

### 2.3. Biofilms

Biofilms are microbial communities that grow on surfaces [48]. A key characteristic of biofilms is that the component microorganisms are encased in an organic matrix, consisting of a variety of

extracellular polymeric substances (EPS) produced by the microorganisms, as well as other organic and inorganic compounds that may be present in the environment [49]. EPS components within the matrix include capsule polysaccharides, extracellular DNA, a variety of secreted proteins, lipids, and frequently OMVs [38]. The EPS composition varies with the microbial community composition as well as the physiology of the component organisms [50]. In the urinary tract, a variety of host-produced organic molecules would also be incorporated into the matrix [16]. One feature of biofilm growth is the generation of microenvironments within the biofilm community (reviewed in [51]). An in vitro study using video microscopy [52] showed how *P. mirabilis* biofilms, and associated urease activity, could protect struvite from dissolution when exposed to slightly acidic (pH 5.8) artificial urine. Crystals not associated with biofilms dissolved readily in this study. Another feature associated with biofilms is the considerable antimicrobial tolerance of the component microorganisms, first described by Nickel et al. [53]. When compared to suspended (planktonic) populations, growth within biofilms may reduce antimicrobial susceptibility by up to three orders of magnitude [54]. Once bacteria leave biofilms and return to a planktonic phase, antimicrobial susceptibility returns, so the term tolerance is more accurate rather than resistance (which implies a permanent heritable change) [55].

As mentioned previously, several studies have shown that struvite calculi are intertwined with biofilms [22,23,26]. Surgical options that are used with struvite calculi include the application of focused sound waves (lithotripsy) to break up the large calculi and allow the voiding of small sand-like calculus remnants with urine [56,57]; or surgical options that are sometimes coupled with antibiotics or urease inhibitors [57]. At least one study showed that various types of lithotripsy caused only a modest loss of viability of *P. mirabilis* within struvite calculi (initial concentration $10^8$ CFU/mL, final concentration $10^4$–$10^6$ CFU/mL after treatment) [58]. Biofilm growth of *P. mirabilis* and other organisms within struvite would greatly decrease their susceptibility to antimicrobial treatments. As well, calculus fragments, which contain viable bacteria, remaining in patients following medical intervention can act as "seeds" for the regrowth of these calculi.

## 3. Struvite Crystal Growth as Biomineralization

Minerals including struvite form as a result of supersaturation and the electrostatic interactions of their component ions [59]. Although bacterial urease [10] and its resulting production of ammonium ($NH_4^+$) and alkaline pH (addressed earlier) are a key factor, the decreased solubility of $Mg^{2+}$ and its attraction towards the other components of struvite, notably ammonium and phosphate, are key. Several molecules have been identified as the therapeutic options for management of struvite urolithiasis [57]. Two examples include acetohydroxamic acid (AHA) and citrate. AHA is a competitive inhibitor of urease, and so acts to lessen the underlying enzymatic mechanism for struvite formation [60]. Citrate can chelate $Mg^{2+}$ and other divalent ions including $Ca^{2+}$ and so reduces the availability of these ions for precipitation [61]. Other inhibitors (reviewed in [62]) may interfere with nucleation or crystal aggregation. A variety of current treatment options are reviewed by Zisman [57].

Bacterial cell surfaces are also a factor in mineral formation (reviewed in [63,64]). Cell surfaces are typically anionic, primarily due to carboxylate (R-COO⁻) and phosphate residues in various components such as membranes, anionic surface proteins, and peptidoglycan. Terminal amino residues (R-NH$_2$) in various cell surface molecules tend to become protonated (R-NH$_3^+$) and where present will give a cationic feature. The chemistry and structure of capsule polysaccharides and LPS O-antigens, which are key components of biofilm matrices, is bacterial strain-specific [65] and can vary widely within a species.

Capsule polymers and other EPS components are often anionic [66] and are typically depicted as amorphous structures when observed by electron microscopy [67], or by their primary chemical structure when analyzed biochemically (e.g., [68]). However, there is evidence that capsule polymers can adopt a secondary structure in the presence of some metal cations, and that this structure may be altered by the ions present [69]. Secondary structure would influence the charge orientation and coordination chemistry of polymers and may influence their potential to influence mineral formation.

Certainly, work would be needed to investigate this hypothesis. One analogy to this concept would be the structure of the ice nucleating protein, present on the cell surfaces of a number of plant pathogens. In its native conformation, the ice-nucleating protein will adopt an orientation that aligns with and promotes ice crystal formation at temperatures slightly below freezing ($-2$ °C). These crystals are a component of frost, which damage plant cells and allow the plant pathogen access to nutrients within the cells. When the structure of the ice-nucleating protein is altered via the mutagenesis of its gene, then the protein structure is altered and the changed protein either has no influence on ice nucleation, or functions as an antifreeze protein [70].

Depending on the growth rate, struvite crystals will adopt characteristic shapes (referred to as crystal habit), ranging from dendritic X-shaped crystals in extremely fast-growing conditions (preferential growth along one crystal axis); to a more prismatic shape (reflecting balanced growth along all crystal axes) [71]. Dumanski et al. observed that struvite crystal growth was enhanced in the presence of a partially anionic capsule polymer from *P. mirabilis* ATCC 49565, but not by the other capsule polymers tested [72]. Interestingly, in that strain of *P. mirabilis*, the capsule polymer has the same structure as O-antigen [68] although it extends at least 0.5 μm from the cell surface, rather than the short distance (~10 nm) seen with other O-antigens [43]. It remains to be seen whether crystal growth enhancement is unique to that strain of *P. mirabilis* or whether it is a general species characteristic.

Recent work by Prywer and colleagues has shown the close association of *P. mirabilis* with struvite [26,73], and has proposed that one feature of *P. mirabilis* struvite formation involves the bacteria aggregating the minerals together based on the cellular zeta potential [74]. Interestingly, Prywer et al. found that carbonate-apatite was aggregated more readily than struvite, although in clinical situations, struvite is the dominant mineral [2]. When measured in suspension, the zeta potential ($\zeta$) can be taken as a parameter to evaluate the dispersion stability of particles. "As a rule, particles with a value of $\zeta \geq \pm 30$ mV can be considered stable. The zeta potential for bacteria alone is close to $-30$ mV and therefore bacteria do not aggregate easily. On the other hand, struvite and carbonate apatite have a zeta potential of several mV and therefore aggregate easily despite the negative value of this potential. For such low zeta potential values, the particles are no longer stable with regard to aggregation. Carbonate apatite aggregates particularly easily because the zeta potential has a value closest to zero" (J. Prywer, personal communication). Bacteria growing within a biofilm would be restricted in mobility by the biofilm matrix. Another issue is that bacteria replicate by binary fission and following the division process, daughter cells may remain attached [75]. Regardless of the biological effects, the investigations by Prywer and colleagues are certainly in line with bacterial cell surface chemistry, and interactions with the surrounding environment (i.e., urine) and metabolic activities described earlier. It opens up some new and intriguing issues related to biomineralization in general and struvite formation in particular.

Bacterial mineralization has also been described in other aspects, including concrete repair and microbial mineral recovery. Certainly, cell surface characteristics including the associated zeta potential of nucleating microorganisms play a role, although specific bacterial metabolic activities are also involved. The interested reader is referred to the representative articles that describe these issues [76,77].

*OMV Influences on Struvite Crystal Growth*

One notable feature of biofilms is the presence of OMVs [38]. Certainly, phospholipid vesicles have been proposed as a model system for influencing biomineralization [78]. While OMVs would contain LPS rather than phospholipids, these molecules are similar in that anionic phosphate moieties are present on the hydrophilic outer surface. The internal components of OMVs include a variety of proteins, lipids and even nucleic acids (reviewed in [39]). The difference in OMV chemistry from one organism to another would arise from their LPS chemistry and the other associated molecules present. In Figure 1, we show a micrograph of a polymicrobial biofilm (Figure 1a) and present a concept in which the OMVs from some bacteria but not others (Figure 1b) may potentially influence struvite growth. In this concept, OMVs with crystal-promoting chemistry would enhance crystal

nucleation and growth, whereas different OMVs with crystal-inhibiting structures would interfere with struvite nucleation. In terms of morphology, OMVs strongly resemble nanobacteria, and so testing the hypothesis that nano-sized objects are involved in biomineralization is potentially significant. Interestingly, there are two publications that suggest that OMVs may promote the formation of calcium oxalate minerals [79,80], a component of metabolic urinary calculi, so in this context the role of nano-sized OMVs in struvite formation is certainly plausible. Unlike bacterial cell walls, in which the shape is maintained by peptidoglycan [65], OMV morphology would likely be influenced by associated mineral formation and solution chemistry.

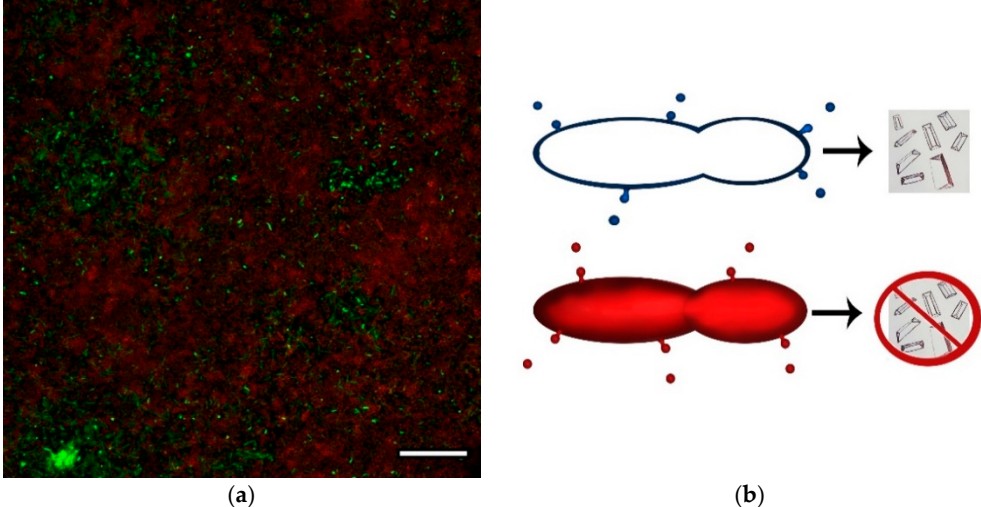

| (a) | (b) |
|-----|-----|

**Figure 1.** Confocal image of a mixed *E. coli* (red-fluorescing) and *P. aeruginosa* (green-fluorescing) biofilm grown in artificial urine (**a**) (Thornhill et al., unpublished). In (**b**), the outer membrane vesicles (OMVs) produced by one organism (blue outline) may promote struvite growth, whereas the OMVs having a different chemistry from another organism (red) may inhibit crystallization. Given the polymicrobial nature of most biofilms, bacterial interactions of cell surface structures including OMVs must be considered during biomineralization studies. Bar in (**a**) represents 30 μm. Micrograph in (**a**) is provided courtesy of S.G. Thornhill.

## 4. Conclusions

Struvite biomineralization is a consequence of complicated urinary tract infections by *P. mirabilis* and other urease-producing bacteria. While urease is well established as a mechanism for struvite precipitation in urine, other factors are also involved. One underappreciated concept is the potential role of cell surface chemistry and the structures including outer membrane vesicles in crystal nucleation, aggregation, and growth. Struvite biomineralization also represents a unique experimental system whereby different factors influencing bacterial-mediated crystal growth can be addressed.

**Author Contributions:** This manuscript was written jointly by R.J.C.M. and E.T.B. All authors have read and agreed to the published version of the manuscript.

**Funding:** Work in R.J.C.M.'s lab related to struvite formation and biomineralization was supported by grants from the Kidney Foundation of Canada, the Natural Sciences and Engineering Research Council of Canada, and an Advanced Research Program grant from the Texas Higher Education Coordinating Board. Current research in R.J.C.M.'s lab is supported by a fellowship from the Prince Foundation and a grant from NASA (NNX17AC79G).

**Acknowledgments:** We thank Starla Thornhill for the use of her confocal micrograph. We also thank several anonymous reviewers for helpful suggestions and Jolanta Prywer for her insightful comments.

**Conflicts of Interest:** The authors declare no conflict of interest. The funders had no role in the design of the study; in the collection, analyses, or interpretation of data; in the writing of the manuscript, or in the decision to publish the results.

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
