# Peer review of "Potential Influences of Bacterial Cell Surfaces and Nano-Sized Cell Fragments on Struvite Biomineralization"

_crystals, doi:10.3390/cryst10080706_

Round 1
Reviewer 1 Report
The review can be accepted in its current form.
Author Response
We thank reviewer one for his/her helpful comments.
Reviewer 2 Report
This brief review is a nice introduction to struvite precipitation in urinary tract infections. It also introduces the novel idea of OMVs as potential nucleation sites for struvite growth.
Line 58. This statement “The cations, Mg2+ and Ca2+ are insoluble at alkaline pH and given the chemical composition of urine, precipitate as struvite (MgNH4PO4.6H2O) and carbonate-apatite (Ca10(PO4)6.CO3) respectively.” should be reworded: “Given the chemical composition of urine, Mg2+ precipitates as struvite (MgNH4PO4.6H2O) and Ca2+ precipitates as carbonate-apatite (Ca10(PO4)6.CO3), both of which are poorly soluble under alkaline conditions.”
Lines 61-63 do not flow well into the rest of that paragraph. Please tie the idea of other genes (non-ureC) to struvite and biofilms.
Line 71. Add a comma after P. mirabilis.
Line 72. Change semi-colon to a comma to avoid sentence fragment.
Line 74. Add comma after cell wall.
Lines 78-79. It is unclear if the close association is between multiple struvite crystals or if the close association is between the crystals and the cells. Please clarify.
Line 89. SEM acronym was previously defined, so no need to define again here.
Line 100. Unclear if study (24) noted that their DNA was a common PCR contaminant or if the authors of the current paper are noting that. Clarify.
Lines 105, 121, 128. Capitalize Gram negative to be consistent with capitalization earlier in manuscript.
Section 2.2. It would be helpful if the authors would explain the purpose of OMV formation. Why do cells produce these? What advantage do they provide?
Line 122. The meaning of “prominent” is unclear here. Does it mean that they are large or numerous? Please clarify.
Line 124. I think that QS is commonly defined as quorum sensing, rather than signaling.
Line 136. The meaning of “growth stage” should be clarified.
Line 139. The acid lability of struvite was already stated on line 60.
Line 162. Add comma after phosphate.
Line 182. This phrase “and may influence mineral formation” needs additional explanation.
Section 3.1 How commonly are OMVs produced? For instance, in a P.aeruginosa biofilm grown in urine/simulated urine, would you expect to see 10 OMVs for every one cell (or some other order of magnitude)?
In Fig 1b, change the red organism to a different color. (My natural tendency was to think that the red organism in 1b was the same as the red organism in 1a.)
Lines 211-212. What are some examples of OMVs with non-crystal-promoting chemistries?
While bacterial cells have reinforcement (i.e., cell wall) to retain their shape, do the authors need to consider whether OMVs would be able to retain their shape once dense precipitates start to form?
The authors might consider including some references from the biomineralization in cement literature. Several authors have determined the (strongly negative) zeta potential of other urease-producing cell surfaces and proposed them as nucleation sites.
Author Response
We thank the reviewer for his/her helpful comments. We have clarified the text and figures in response to the helpful comments.
We addressed the grammatical corrections in the text as suggested.
For the association between struvite crystals and cells, during in vitro observations of the initial stages (e.g. Infect Immun doi: 10.1128/IAI.49.3.805-811.1985) the crystals are smaller than cells, whereas in mature calculi, there may be one or more larger crystals in proximity to bacterial cells (e.g. J. Urol. doi: 10.1016/s0022-5347(17)49116-6.; J. Med. Microbiol. doi: 10.1099/00222615-29-1-1. These articles have all been cited.
With regards to the Kajander article (reference 24) and the critique in the Cisar et al paper (both published in PNAS), my understanding was that Kajander and Ciftcioglu were trying to extract DNA from the nanobacteria. There is no mention of containment issues (i.e. use of a biosafety cabinet) to control PCR contamination in this paper, so the potential for contamination does exist. We tried to clarify the text.
The word "Gram" has been capitalized.
In Section 2.2, we added a short description of OMV formation. There are several reviews that address this and they were cited.
We appreciate the reviewer's suggestion for modifying our figure and have done so to improve clarity.
Our hypothesis of crystal-promoting chemistry of OMVs is currently a hypothesis and would need to be tested, so included the phrase (may influence mineral formation). We were informed by another reviewer that OMVs have been shown to promote calcium oxalate crystal growth and have included appropriate references. With respect to crystal-promoting and crystal-non promoting, two ideas come to mind. The first is the specific role of some but not other capsule polymers (Dumanski et al paper). A second is the ability of bacterial surface proteins to nucleate ice crystals, and to lose this ability if mutated. We have added 1-2 sentences to clarify these issues.
We replaced the phrase "growth stage" with "physiology" as that term better explains the concept of that sentence.
To our knowledge, no studies have been performed to measure OMV formation in urine or artificial urine.
We did mention that OMV morphology may change as a result of mineral formation as there is no peptidoglycan associated.
Finally, we did add a representative references showing the role of negative zeta potential of organisms as nucleation sites during concrete formation and also microbial mineral recovery.
Reviewer 3 Report
In the article entitled "Potential influences of bacterial cell surfaces and nano-sized cell fragments on struvite biomineralization" by McLean et al., the information about the influence of the bacterial components on the in struvite crystallization was presented. Struvite is one of the mineral components of urinary stones formed as a result of infection with urease-positive bacteria, e.g. Proteus mirabilis. While the role of urease in this process is well known, the role of other bacterial factors is still under investigation. Their participation seems to be very important as they modulate the intensity of crystallization, influence the aggregation of crystals and provide nuclei of crystallization and this aspect is addressed in this work. In this review, the authors briefly and concisely describe the participation of bacterial surface components, the EPS of the biofilm, and nano-sized objects including nanobacteria and OMVs in the biocrystallization. I consider the inclusion of OMV’s in this work as the most interesting part. However, the manuscript needs some improvement, below are my remarks and comments to the authors.
- the authors provide information that nano-sized objects/nanobacteria „have been described in association with urinary calculi including struvite calculi” (abstract; line 15-16) but in the text of the manuscript this information is no longer given or described in more detail with the reference to the literature source, which requires clarification or supplementation
- the names of Gram-positive and Gram-negative bacteria are given in different ways, e.g. Gram negative (line 114) but below the text - gram negative line 121; this should be standardized throughout the text
- line 71 page 2 “ …P. mirabilis a gram negative member of the family Enterobacteriaceae…” According to data published by Adeolu et al. Proteus mirabilis, in line with changes in taxonomy, currently does not belong to the Enterobacteriaceae family, but to the Morganellaceae (Adeolu M et al Int J Syst Evol Microbiol 66: 5575-5599. http://doi: 10.1099/ijsem.0.001485)
- line 38-39 page 1,” ..struvite urinary calculi account for approximately 10-15% of all urinary calculi..” this information is supported by quite an old reference, is this percentage the same at the moment?
- Figure 1b shows two cells of different colors producing outer membrane vesicles, one of them, according to the description, promotes crystallization while the other one inhibits the process however the only obvious difference is that in color. It would be worth supplementing this figure by adding information about other differences between cells e.g. features, mechanism of action etc.
- Although there are no data on the participation of OMV’s in struvite crystallization so far, their participation has been proven in the crystallization of calcium oxalate which is a component of metabolic urinary stones.( Amimanan, P. et al. (2017). Sci. Rep. 7:2953. doi: 10.1038/s41598-017-03213-x; Kanlaya R, et al. (2019) Front. Microbiol. 10:2507. doi: 10.3389/fmicb.2019.02507). These data do not apply to struvite, however, due to the fact that it proves participation of OMV’s in biomineralization it would be worth adding them to the text.
Author Response
We thank the reviewer for helpful comments and suggestions.
The nano-sized objects referred to in the abstract are addressed in section 2.1.
We have tried to standardize bacterial names.
We thank the reviewer for the correction of Proteus genus as now being in Morganellaceae and have added the reference provided.
The incidence of struvite urinary calculi according to a 2017 review by Das et al https://doi.org/10.1016/j.biopha.2017.10.015 is 12%, which is similar to the range (10-15%) quoted in the text. We added the more recent review citation.
We have revised Figure 1 to provide more clarity.
Finally, we greatly appreciate the reviewer's suggestion for OMV participation in calcium oxalate calculi and have included the appropriate text. We missed that important work and are grateful to the reviewer for alerting us.
Reviewer 4 Report
Major Comments:
The authors review the literature on the role of bacterial surfaces and outer membrane vesicles in promoting struvite calculi formation in the urinary tract. The authors note biofilms may protect struvite from acidic pHs where they would normally dissolve. Further, they review the earlier idea of so-called “nanobacteria” and suggest that these are probably either outer membrane vesicles (OMVs) of Gram negative bacteria or other cellular debris. They support the more recent studies which suggest that the lipolysaccharide component of OMVs may be responsible for promoting struvite formation. They also propose that the ability of OMVs from different bacterial species to promote struvite formation or not, may depend on the detailed chemical compositional variations in the LPS of different bacteria. The manuscript is written clearly and provides a good overview of the literature. I recommend accepting for publication with minor modifications.
Minor Comments:
Line 159 and 162: Change “chemical attraction” to “electrostatic interactions.”
Line 195. Prywer et al. (2015) shows negative zeta potential for cell surface, struvite and HAP, with the negative value increasing as HAP < Struvite < Cell surface. How do two negatively charged surfaces aggregate? Please comment.
Author Response
We thank the reviewer for his/her helpful comments.
We have changed the "chemical attraction" to "electrostatic interactions"
With respect to the zeta potential question, we have added some text in lines 214-220 to explain the issue. As well, the particles are not suspended in solution but would be mixed with other ions (e.g. Mg 2+ and Ca 2+, which could serve as bridging ions between anionic particles); and in the case of a biofilm-associated mineral, would also interact with the organic biofilm matrix that would restrict dispersion. I hope our explanation will clarify this issue.